# Population-based identification and temporal trend of children with primary nephrotic syndrome: The Kaiser Permanente nephrotic syndrome study

Rishi V. Parikh[1], Thida C. Tan[1], Dongjie Fan[1], David Law[2], Anne S. Salyer[2], Leonid Yankulin[3], Janet M. Wojcicki[4,5], Sijie Zheng[2], Juan D. Ordonez[2], Glenn M. Chertow[5,6], Farzien Khoshniat-Rad[1], Jingrong Yang[1], Alan S. Go[1,5,6,7]*

1 Division of Research, Kaiser Permanente Northern California, Oakland, CA, United States of America, 2 Department of Nephrology, Kaiser Permanente Oakland Medical Center, Oakland, CA, United States of America, 3 Department of Nephrology, Kaiser Permanente San Francisco Medical Center, San Francisco, CA, United States of America, 4 Department of Pediatrics, University of California, San Francisco, San Francisco, CA, United States of America, 5 Departments of Medicine and Epidemiology and Biostatistics, University of California, San Francisco, San Francisco, CA, United States of America, 6 Departments of Medicine (Nephrology) and Epidemiology and Population Health, Stanford University School of Medicine, Stanford, CA, United States of America, 7 Department of Health System Science, Kaiser Permanente Bernard J. Tyson School of Medicine, Pasadena, CA, United States of America

* Alan.S.Go@kp.org

**Data Availability Statement:** The minimal de-identified data set necessary to replicate the analyses is in the supporting information files.

## Abstract

### Introduction

Limited population-based data exist about children with primary nephrotic syndrome (NS).

### Methods

We identified a cohort of children with primary NS receiving care in Kaiser Permanente Northern California, an integrated healthcare delivery system caring for >750,000 children. We identified all children <18 years between 1996 and 2012 who had nephrotic range proteinuria (urine ACR>3500 mg/g, urine PCR>3.5 mg/mg, 24-hour urine protein>3500 mg or urine dipstick>300 mg/dL) in laboratory databases or a diagnosis of NS in electronic health records. Nephrologists reviewed health records for clinical presentation and laboratory and biopsy results to confirm primary NS.

### Results

Among 365 cases of confirmed NS, 179 had confirmed primary NS attributed to presumed minimal change disease (MCD) (72%), focal segmental glomerulosclerosis (FSGS) (23%) or membranous nephropathy (MN) (5%). The overall incidence of primary NS was 1.47 (95% Confidence Interval:1.27–1.70) per 100,000 person-years. Biopsy data were available in 40% of cases. Median age for patients with primary NS was 6.9 (interquartile range:3.7 to 12.9) years, 43% were female and 26% were white, 13% black, 17% Asian/Pacific Islander, and 32% Hispanic.

**Funding:** This study was funded by the Brin Wojcicki Foundation. ASG declares funding from the Brin Wojcicki Foundation (grant number not applicable). The funders had no role in study design, data collection and analysis, decision to publish, or preparation of the manuscript.

**Competing interests:** I have read the journal's policy and the authors of this manuscript have the following disclosures: ASG declares funding from the Brin Wojcicki Foundation. GMC declares grants from NIDDK, and Amgen, and fees from Ardelyx, AstraZeneca, Baxter, Cricket, DiaMedica, Gilead, Reata, Sanifit, Vertex, Satellite Healthcare, Angion, Bayer, ReCor and other disclosures from CloudCath, Durect, and Outset. All other authors have nothing to disclose. This does not alter our adherence to PLOS ONE policies on sharing data and materials.

## Conclusion

This population-based identification of children with primary NS leveraging electronic health records can provide a unique approach and platform for describing the natural history of NS and identifying determinants of outcomes in children with primary NS.

## Introduction

Nephrotic syndrome (NS) is one of the most common kidney disorders in children, with an estimated population incidence ranging from 2 to 16 cases per 100,000 children depending on the setting and population studied [1]. Although children may exhibit NS secondary to other diseases, medication use, or infections, most pediatric NS is considered idiopathic and has three main general attributed etiologies: minimal change disease (MCD), focal segmental glomerulosclerosis (FSGS), and membranous nephropathy (MN) [1, 2]. Children with NS are at risk for various potential adverse short- and long-term outcomes, including higher rates of infection, hypertension, venous thromboembolism, fractures, and progression to chronic kidney disease and end-stage kidney disease [3–8]. Therefore, appropriate systematic identification and population management strategies for children with NS, especially primary NS, are needed to facilitate prevention of subsequent clinical complications.

Valuable insights on the etiology, management strategies, and outcomes of NS have come primarily from selected prospective cohort studies or registries for children with NS [9–16]. However, these studies often require biopsy-confirmed NS for inclusion, are limited to patients identified with diagnosis codes for NS, rely on data only from tertiary care medical centers or other selected referral settings, or combine all types of NS, such that results may not fully reflect the population-level burden of primary NS.

In this study, we estimate and characterize the population incidence of pediatric primary NS in a large, integrated healthcare delivery system through structured review of available data within electronic health records (EHR) and associated health system data sources.

## Methods

### Source population and study sample

The source population included members of Kaiser Permanente Northern California (KPNC), an integrated health care delivery system currently providing comprehensive care to >4.5 million members throughout Northern California. Its membership is highly representative of the regional and statewide population with regard to sociodemographic characteristics [17].

The study sample included all pediatric (age <18 years) health plan members who had nephrotic range proteinuria and/or a diagnosis code suggestive of possible NS between January 1, 1996 and December 31, 2012 using methods described in detail below. Eligible patients were identified for this study using laboratory test results and diagnosis codes associated with clinical encounters. After identifying the initial cohort of eligible patients based on EHR algorithms, targeted subgroups were selected for manual adjudication of medical records by board-certified nephrologists to confirm the presence of NS, and assign the type of NS (primary vs. secondary) and presumed cause of primary NS (MCD, FSGS or MN).

The study was approved by the KPNC institutional review board, and waiver of informed consent was obtained due to the nature of this retrospective data-only study.

## Identification of potential nephrotic syndrome using proteinuria results

We identified children with nephrotic range proteinuria if they had ≥1 positive urine test result from any setting (inpatient or outpatient) found in health system laboratory databases using any of the following criteria: albumin-to-creatinine ratio (ACR) >3500 mg/g; protein-to-creatinine ratio (PCR) >3.5 mg/mg; 24-hour urine protein excretion >3500 mg; or ≥3+ on urine protein dipstick. These criteria are designed to be more specific for electronic ascertainment of potential nephrotic syndrome using thresholds of proteinuria values that are more stringent than typical clinical cutoffs [18].

## Identification of potential nephrotic syndrome using diagnosis codes

We included children in the cohort if they had ≥1 primary discharge, outpatient, or emergency department diagnosis of NS during our sampling timeframe. Diagnoses of NS were ascertained from health plan databases based on relevant *International Classification of Diseases*, *Ninth Edition* (ICD-9) codes. The following ICD-9 codes were used for identification: 581, 581.0, 581.1, 581.2, 581.3, 581.8, 581.81, 581.89, and 581.9. The diagnosis codes were initially categorized into groups for facilitating the targeted physician adjudication process. Specifically, manual physician review of health records was prioritized for (1) children with documented nephrotic range proteinuria and a diagnosis of NS, (2) children with a diagnosis of NS only, and (3) children with documented nephrotic range proteinuria in the absence of a diagnosis of NS.

## Validation of nephrotic syndrome and assembly of final cohort

The overall cohort that underwent validation included 2,541 children identified with a qualifying proteinuria measurement or diagnosis codes of NS in the absence of diabetes mellitus, with the following specific validation groups: any NS diagnosis (N = 1,281); nephrotic range proteinuria based on laboratory measurements of urine ACR, PCR, or 24-hour urine protein and no documented diagnosis of kidney disease (N = 85); nephrotic range proteinuria based on ≥3 urine dipstick measurement of 3+ proteinuria and no documented diagnosis of kidney disease (N = 329); proteinuria based on ≥1 qualifying urine dipstick measurement and receiving ≥1 medical therapy used for NS (i.e., angiotensin-converting enzyme inhibitors, angiotensin II receptor blockers, azathioprine, cyclosporine, cyclophosphamide, methylprednisone, prednisone, prednisolone, tacrolimus, and mycophenolate mofetil) within 1 year of index date (N = 746); and nephrotic range proteinuria based on two urine dipstick measurements of 3 + proteinuria and no documented diagnosis of kidney disease. For the last criterion, a random sample of 100 patients was reviewed given the very large number of identified patients to evaluate the potential yield for identifying primary NS.

Children confirmed with NS using data from the EHR and manual review of medical records required evidence of symptoms and/or signs consistent with NS and a laboratory measurement indicating nephrotic range proteinuria at the time of diagnosis. Presumed cause of NS was ascertained by review of biopsy results in KPNC pathology databases, if available. For children without available biopsy results in KPNC, we incorporated information from nephrology or other treating physician notes, other laboratory values, and treatment patterns to assign the presumed etiology. Biopsies conducted outside of KPNC or before joining KPNC were not routinely captured in pathology databases and were reviewed only through manual review of provider notes. All causes of NS were considered presumed, unless a definitive biopsy result was available within or outside KPNC. More details on the specific methodology for adjudication of medical records, exclusion criteria, and rules are described in the **S1 File**. We excluded patients if the nephrologist reviewer could not confirm a diagnosis of NS using

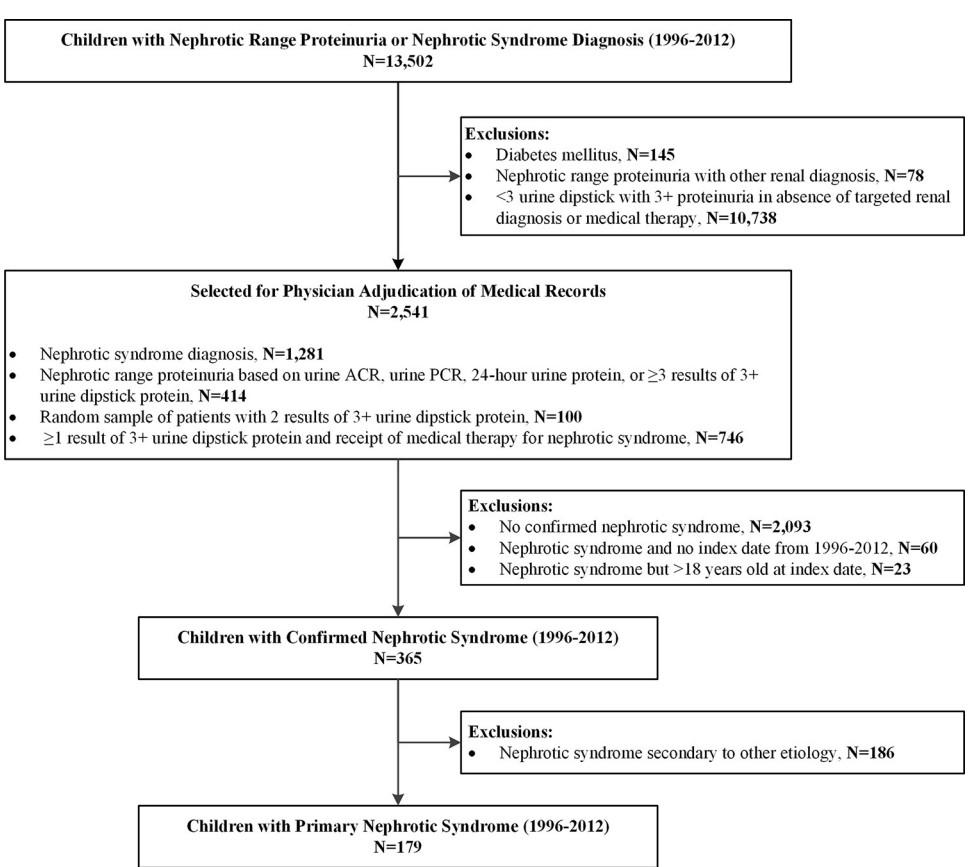

**Fig 1. Cohort assembly of children with nephrotic syndrome between January 1, 1996 and December 31, 2012.**

the criteria described above, or among confirmed NS cases, if we could not establish an index date (i.e., date of initial NS diagnosis), if the index date was before 1996 or after 2012, or if the patient was older than 18 years based on the index date found. The overall assembled cohort included 365 children with confirmed NS (**Fig 1**).

## Covariates

Demographic characteristics (age, gender and self-reported race/ethnicity, if available) were obtained from health plan databases. We defined targeted comorbidities by diagnosis or procedure codes supplemented with available laboratory test results, outpatient vital sign data, and/or prescribed medications using EHR-based data that were cleaned and linked at the individual-patient level into the KPNC Virtual Data Warehouse as previously described and validated [19, 20]. Household educational attainment and annual income were estimated using residential block-level information from U.S. census data. Low education was defined as living in a census block where more than 25% of those aged 25 years or older had less than a 12th-grade education; low income was defined as living in a block where median annual household income was less than $35,000 per year. Patient vital status was determined using comprehensive information from health plan administrative and clinical databases, member proxy reporting, Social Security Administration vital status files, and California state death certificate information.

## Statistical analyses

We conducted all analyses using SAS version 9.3 (Cary, N.C.). We first characterized the overall cohort based on the presumed etiology of confirmed NS. Next, given the focus of the study on primary NS, we calculated crude incidence (per 100,000 person-years) and associated 95% confidence limits of primary NS during the study period by dividing the number of confirmed primary NS cases by the total person-years contributed by the pediatric KPNC population with no diabetes mellitus. We similarly calculated the crude annual incidence (per 100,000 person-years) and associated 95% confidence intervals of primary NS due to presumed MCD; the relatively small number of cases of either presumed FSGS or MN precluded our reliably calculating incidence of for those types of primary NS. We averaged annual rates into the following categories of time (1996–1999, 2000–2003, 2004–2006, 2007–2009 and 2010–2012) for increased precision per time period. We calculated the age- and sex-adjusted incidence and associated 95% confidence limits of primary NS per category of time, directly standardized to the 2010–2012 population, as well as the age- and sex-stratified crude incidence of primary NS. We performed Cochrane-Armitage tests to evaluate for significant linear trends over time.

## Results

### Cohort assembly

Among 2,541 eligible children with potential NS whose medical records were manually reviewed, we confirmed NS in 365 children (179 [47%] classified with primary NS), and the proportion of confirmed NS cases varied substantially by method of EHR-based identification (**Table 1**). Children with documented proteinuria and a diagnosis code of NS were confirmed to have NS in 39% of cases as compared to only 6.5% of children who had a diagnosis code of NS but no documented proteinuria found in health system laboratory databases.

Among 179 children with primary NS, we classified 129 (72%), 42 (23%), and 8 (5%) presumably attributed to MCD, FSGS, and MN, respectively. Of the remaining 186 children with secondary NS, 46 (25%) had other/unspecified glomerulonephritis, 39 (21%) IgA nephropathy, 16 (9%) membranoproliferative glomerulonephritis, and 92 (49%) with other causes or no clearly identified cause (**Fig 2**).

### Baseline characteristics

Among children with primary NS, median age was 6.9 (interquartile range: 3.7 to 12.9) years, 43% were female, 26% were White, 13% were Black, 17% were Asian/Pacific Islander, and 32% identified as Hispanic (**Table 2**). Of note, one third of patients lived in a household with low educational attainment and 15% of patients in households with a median household income of less than $35,000 based on census-based classification during the study period. The prevalence of diagnosed chronic lung disease was 22%, while other documented comorbid conditions were infrequent. Data on body mass index, blood pressure, serum albumin, lipoproteins and hemoglobin were unavailable in a significant fraction of patients (**Table 2**).

Characteristics varied by presumed etiology of primary NS. Compared to those with presumed FSGS, children with presumed MCD were younger and had higher eGFR at index date. Although 30% of children had unknown self-reported race, children with FSGS were more likely to be Black or multiracial than patients with other etiologies of primary NS, and children with MCD were more likely to be of Asian or Pacific Islander descent. As noted previously, a large proportion of all patients did not have available body mass index, blood pressure and selected laboratory tests, which precluded comparison of those measures across types of primary NS. Kidney biopsies were performed at KPNC in 72 (40%) children classified with

**Table 1. Proportion of children with confirmed nephrotic syndrome, stratified by identification approach.**

| Identification Approach | Total Charts Reviewed | Confirmed Nephrotic Syndrome N (%) |
|---|---|---|
| Nephrotic syndrome diagnosis code with MCD*, FSGS† or MN‡ diagnosis code and documented proteinuria | 33 | 21 (63.6) |
| Nephrotic syndrome diagnosis code with MCD, FSGS or MN diagnosis code and no documented proteinuria | 9 | 2 (22.2) |
| Nephrotic syndrome diagnosis code with diagnosis code for other cause and documented proteinuria | 434 | 126 (29.0) |
| Nephrotic syndrome diagnosis code with diagnosis code for other cause and no documented proteinuria | 415 | 24 (5.8) |
| MCD, FSGS or MN diagnosis code alone and documented proteinuria | 35 | 16 (45.7) |
| MCD, FSGS or MN diagnosis code alone and no documented proteinuria | 6 | 0 (0.0) |
| Nephrotic syndrome diagnosis code with other listed cause and documented proteinuria | 243 | 127 (52.3) |
| Nephrotic syndrome diagnosis code with other listed cause and no documented proteinuria | 106 | 9 (8.5) |
| No renal diagnosis but with documented proteinuria from urine albumin-to-creatinine ratio, protein-to-creatinine ratio, or 24-hour urine protein excretion measurements | 85 | 8 (9.4) |
| No renal diagnosis but with ≥3 results of 3+ urine dipstick proteinuria | 329 | 16 (4.8) |
| No renal diagnosis but with ≥1 urine dipstick proteinuria and ≥1 therapy used for nephrotic syndrome | 746 | 16 (2.1) |
| Random sample of patients with no renal diagnosis but with 2 results of 3+ urine dipstick proteinuria | 100 | 0 (0.0) |

*MCD = minimal change disease

† FSGS = focal segmental glomerulosclerosis

‡ MN = membranous nephropathy

primary NS, with a higher biopsy rate in those with FSGS (86%) and MN (75%) than in those cases attributed to MCD (23%) (**Table 2**).

## Temporal trends of primary NS

The overall incidence of primary NS within our pediatric population was 1.47 (95% Confidence Interval [CI]:1.27–1.70) per 100,000 person-years. The incidence of all primary NS increased significantly over the study period, from 1.04 (95% CI:0.72–1.51) per 100,000 person-years in 1996–1999, to 1.88 (95% CI:1.38–2.55) per 100,000 person-years in 2007–2009 and 1.62 (95% CI:1.17–2.24) per 100,000 person-years in 2010–2012 (**Fig 3**). The incidence of primary NS due to presumed MCD also increased over the study period (P = 0.01 for trend) (**Fig 3**). After standardizing for the distribution in age and sex using the 2010–2012 Kaiser Permanente Northern California overall pediatric population, the adjusted incidence was similar to the crude incidence (**S1 Fig in S1 File**). When stratified by age categories, we observed a significant increase in the incidence of primary NS among children aged 6–11 years, but not among children aged 0–5 years or 12–17 years (**S2 Fig in S1 File**). We also observed a significant increase in the incidence among boys over time, but not girls, with the overall rates being higher in boys than in girls (**S3 Fig in S1 File**).

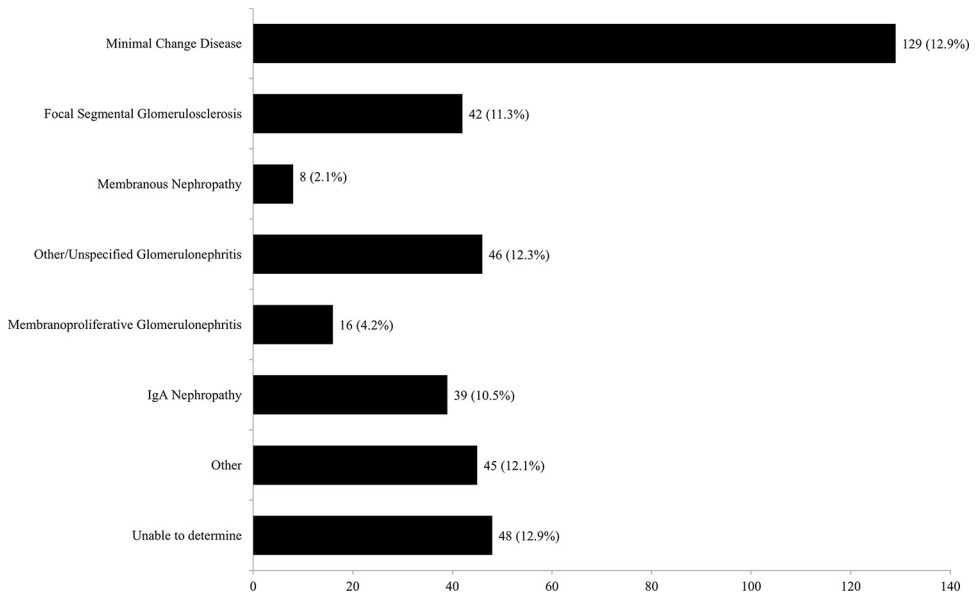

**Fig 2. Distribution of presumed etiology in 365 children with confirmed nephrotic syndrome.**

## Discussion

Leveraging EHR, clinical and administrative data over a 15-year period within a pediatric population receiving care in an integrated healthcare delivery system in California, we developed a structured methodology to systematically delineate the population incidence and selected characteristics of children with primary NS. We identified 179 children who had confirmed primary NS, with an incidence of 1.47 per 100,000 person-years averaged over the entire study period. We also observed that the incidence of confirmed primary NS increased within our pediatric population from 1.04 per 100,000 person-years in 1996–1999 to 1.62 per 100,000 person-years in 2010–2012. This increase may be driven by increases in incidence among children aged 6–11 years, and particularly among boys compared to girls. However, it is unclear whether this increase reflects a true increase in NS incidence in the underlying populations, or a change in practice patterns or population demographics over time. Children with primary NS had low comorbidity burden overall, with the exception of diagnosed chronic lung disease which have been previously described to be associated with certain types of NS in children [21, 22]. As expected, the availability of kidney biopsies in KPNC among children with presumed MCD was much lower than in children classified with FSGS or MN, as invasive procedures are frequently not performed in children with lower severity or steroid-sensitive NS. When applied retrospectively and prospectively, these methods have the potential to aid in the systematic identification of children with NS as well as in designing population-based care delivery strategies as well as facilitating observational research and recruitment into clinical trials.

Our observed incidence estimates are comparable to the limited number of other population-based studies on pediatric NS. In the Netherlands, Bakkali et al. observed an overall NS incidence of 1.52 per 100,000 person-years in children from 2003–2006 but did not distinguish between primary and secondary NS [12]. In a single-center, hospital-based study of diagnosed pediatric NS in Toronto, Banh et al. observed an increase in incidence from 1.99 to 4.71 per 100,000 person-years from 2001 to 2011 [23]. In addition, Kikunaga et al. reported an incidence of 6.49 per 100,000 person-years in Japan from 2010–2012, although this estimated incidence was based on survey results and may overestimate true incidence based on standardized

**Table 2. Characteristics of 179 children with primary nephrotic syndrome due to confirmed minimal change disease, focal segmental glomerulosclerosis or membranous nephropathy identified between 1996–2012.**

| Variable | Overall | Minimal Change Disease | Focal Segmental Glomerulosclerosis | Membranous Nephropathy |
|---|---|---|---|---|
| | (N = 179) | (N = 129) | (N = 42) | (N = 8) |
| **Age, yr** | | | | |
| Median (IQR) | 6.9 (3.7–12.9) | 5.9 (3.6–9.9) | 11.4 (6.3–16.2) | 14.5 (3.3–16.5) |
| Range | 0.8–17.8 | 1.2–17.8 | 0.8–17.8 | 2.1–17.8 |
| **Age group, yr, n (%)** | | | | |
| 0–5 | 78 (43.6) | 65 (50.4) | 10 (23.8) | 3 (37.5) |
| 6–9 | 38 (21.2) | 32 (24.8) | 6 (14.3) | 0 (0.0) |
| 10–15 | 36 (20.1) | 20 (15.5) | 13 (31.0) | 3 (37.5) |
| 16–17 | 27 (15.1) | 12 (9.3) | 13 (31.0) | 2 (25.0) |
| **Female, n (%)** | 77 (43.0) | 58 (45.0) | 16 (38.1) | 3 (37.6) |
| **Race, n (%)** | | | | |
| White | 43 (24.0) | 29 (22.5) | 11 (26.2) | 3 (37.5) |
| Black | 23 (12.8) | 15 (11.6) | 7 (16.7) | 1 (12.5) |
| Asian/Pacific Islander | 31 (17.3) | 29 (22.5) | 2 (4.8) | 0 (0.0) |
| Multiracial | 30 (16.8) | 20 (15.5) | 9 (21.4) | 1 (12.5) |
| American Indian/Alaska Native | 1 (0.6) | 1 (0.8) | 0 (0.0) | 0 (0.0) |
| Unknown | 51 (28.5) | 35 (27.1) | 13 (31.0) | 3 (37.5) |
| **Hispanic ethnicity, n (%)** | 62 (34.6) | 42 (32.6) | 16 (38.1) | 4 (50.0) |
| **Household educational attainment, n (%)** | | | | |
| > 25% with less than 12[th] education | 56 (31.3) | 39 (30.2) | 14 (33.3) | 3 (37.5) |
| ≤ 25% with less than 12[th] education | 117 (65.4) | 85 (65.9) | 28 (66.7) | 4 (50.0) |
| Unknown | 6 (3.4) | 5 (3.9) | 0 (0.0) | 1 (12.5) |
| **Median household income, n (%)** | | | | |
| < $35,000 | 26 (14.5) | 21 (16.3) | 5 (11.9) | 0 (0.0) |
| ≥ $35,000 | 147 (82.1) | 103 (79.8) | 37 (88.1) | 7 (87.5) |
| Unknown | 6 (3.4) | 5 (3.9) | 0 (0.0) | 1 (12.5) |
| **Baseline medical history, n (%)** | | | | |
| Hypertension | 7 (3.9) | 6 (4.7) | 1 (2.4) | 0 (0.0) |
| Dyslipidemia | 7 (3.9) | 7 (5.4) | 0 (0.0) | 0 (0.0) |
| Chronic liver disease | 1 (0.6) | 1 (0.8) | 0 (0.0) | 0 (0.0) |
| Chronic lung disease | 47 (26.3) | 33 (25.6) | 13 (31.0) | 1 (12.5) |
| Hyperthyroidism | 1 (0.6) | 0 (0.0) | 0 (0.0) | 1 (12.5) |
| Hypothyroidism | 3 (1.7) | 2 (1.6) | 0 (0.0) | 1 (12.5) |
| Cancer | 3 (1.7) | 3 (2.3) | 0 (0.0) | 0 (0.0) |
| **Baseline laboratory values, n (%)** | | | | |
| Serum creatinine, mg/dL | | | | |
| Median (IQR) | 0.4 (0.3–0.6) | 0.4 (0.3–0.5) | 0.6 (0.4–0.8) | 0.7 (0.4–0.9) |
| Missing, n(%) | 43 (24.0) | 34 (26.4) | 9 (21.4) | 0 (0.0) |
| Serum albumin, mg/dL | | | | |
| Median (IQR) | 2.2 (1.6–3.3) | 2.0 (1.6–3.2) | 2.9 (2.0–3.8) | 2.0 (2.0–2.6) |
| Missing, n (%) | 58 (32.4) | 44 (34.1) | 13 (31.0) | 1 (12.5) |
| Total cholesterol, mg/dL | | | | |
| <200 | 9 (5.0) | 6 (4.7) | 2 (4.8) | 1 (12.5) |
| 200–240 | 7 (3.9) | 4 (3.1) | 3 (7.1) | 0 (0.0) |
| > 240 | 80 (44.7) | 60 (46.5) | 15 (35.7) | 5 (62.5) |
| Unknown | 83 (46.4) | 59 (45.7) | 22 (52.4) | 2 (25.0) |

(*Continued*)

**Table 2.** (Continued)

| Variable | Overall | Minimal Change Disease | Focal Segmental Glomerulosclerosis | Membranous Nephropathy |
|---|---|---|---|---|
| | (N = 179) | (N = 129) | (N = 42) | (N = 8) |
| Hemoglobin, g/dL | | | | |
| 9.0–9.9 | 2 (1.1) | 1 (0.8) | 1 (2.4) | 0 (0.0) |
| 10.0–10.9 | 8 (4.5) | 4 (3.1) | 3 (7.1) | 1 (12.5) |
| 11.0–11.9 | 14 (7.8) | 10 (7.8) | 2 (4.8) | 2 (25.0) |
| 12.0–12.9 | 32 (17.9) | 22 (17.1) | 9 (21.4) | 1 (12.5) |
| 13.0–13.9 | 29 (16.2) | 23 (17.8) | 4 (9.5) | 2 (25.0) |
| ≥ 14 | 33 (18.4) | 25 (19.4) | 7 (16.7) | 1 (12.5) |
| Unknown | 61 (34.1) | 44 (34.1) | 16 (38.1) | 1 (12.5) |

diagnostic criteria [24]. Lastly, Dossier et. al. observed a much higher incidence of idiopathic NS of 3.35 per 100,000 children in Paris from 2007–2010 [25]. Variation in reported incidence across populations is not fully understood but is likely driven, at least in part, by differences in the identification methodology as well as racial and ethnic distributions of the source populations. For example, the majority of Toronto cohort members were South Asians who are known to have a higher incidence of NS, and the Paris cohort used solely laboratory values to identify NS likely leading to inclusion of secondary NS [23, 25, 26]. In contrast to other studies using only diagnosis codes to identify NS, we additionally performed targeted manual chart review by board-certified nephrologists for evidence of a clinical presentation consistent with NS which enhanced our specificity and reduced misclassification compared with reliance only on administrative diagnosis codes. To our knowledge, only one other published study has attempted to evaluate data from EHR systems to identify rare kidney diseases in a pediatric population. In that study, Denburg et al. developed computable phenotypes to identify a broad

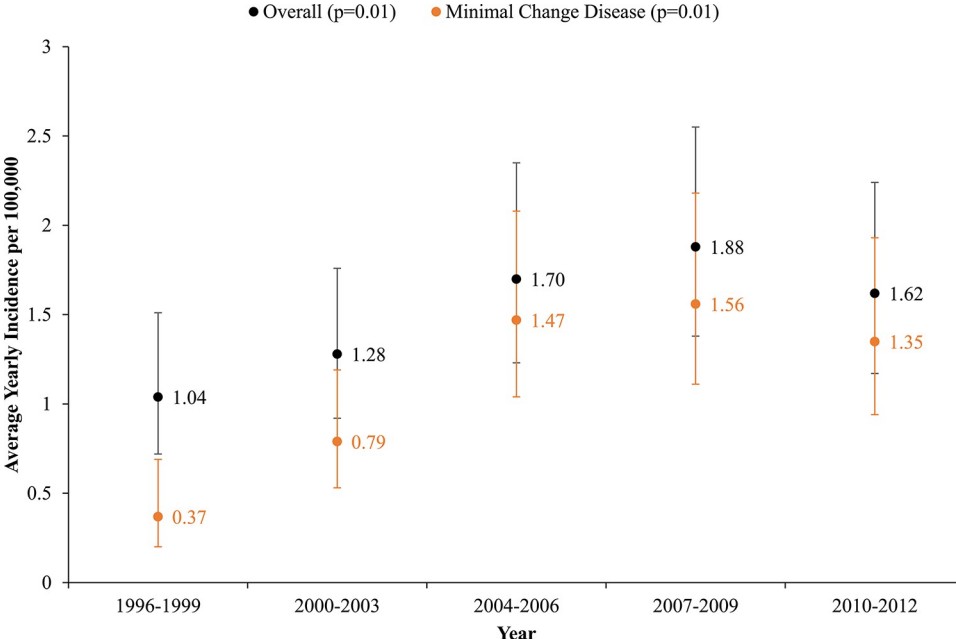

**Fig 3. Average 1-year incidence of primary nephrotic syndrome among all pediatric Kaiser Permanente Northern California members without diabetes between 1996 and 2012, overall and for minimal change disease.**

set of glomerular diseases using a combination of diagnosis codes, transplant procedure codes, and kidney biopsies [27]. While the authors did not incorporate laboratory data or address primary NS, their findings further support the potential of using EHR data to accurately identify patients with glomerular diseases, including idiopathic NS, which can enable more population-based care and research for rarer kidney diseases.

A major strength of our study is the inclusion of a large, ethnically diverse pediatric population receiving care in a fully integrated healthcare delivery system where essentially all care is coordinated across inpatient, emergency department and outpatient settings. This allowed for the assessment of available laboratory data across all practice settings to systematically screen for children with evidence of nephrotic-range proteinuria, as well as the assessment of baseline demographic and risk factors and conditions, if documented in the EHR. In addition, all primary NS cases were manually confirmed through targeted manual medical records review by nephrologists, which is considered the gold standard for retrospective case identification and also demonstrated the suboptimal performance of diagnosis codes alone to identify children with NS.

Our study also has certain limitations. Because this retrospective study relied on data collected as part of clinical care across an earlier time period in children, comprehensive data were not available for all variables of interest, with a significant proportion of patients missing data for vital signs, body mass index and selected laboratory tests. Biopsy reports from procedures performed at other facilities were also not comprehensively available for review, and the presumed etiologies of MN and FSGS in those patients were derived using diagnosis codes and other provider notes. This lack of available laboratory and biopsy data is likely also driven by less severe or steroid sensitive NS cases where a provider may choose not to perform a more invasive blood test or procedure if the proteinuria and clinical presentation is consistent with NS and was controlled after an initial treatment regimen. This is in contrast to prospective studies using a structured research protocol where recruited children with NS can undergo regular data collection and laboratory testing. However, we believe that our approach could be widely used by health systems with available EHR-based laboratory and diagnosis data to facilitate identification of children with potential primary NS who may benefit from systematic evaluation and laboratory testing to confirm their diagnosis and subsequent disease management. We recognize that manual review of medical records by nephrologists to confirm cases of primary NS on a large scale can be time- and resource-intensive, so additional efforts are needed to refine diagnostic approaches in even more contemporary populations and advanced EHR systems, along with incorporating methods such as natural language processing and other machine learning methods applied to unstructured EHR notes and biopsy reports [28]. Given the lower number of children with primary NS due to presumed FSGS and MN, we were unable to calculate stable estimates of changes in incidence of these types of primary NS over the study period. We were also unable to further characterize racial differences in the incidence of primary NS due to the limited availability of data on self-reported race. Our proteinuria thresholds for the initial electronic identification of children with NS were more stringent by design to increase specificity. This may have resulted in underestimation of NS incidence among those without a diagnosis code and less severe proteinuria. Although the large majority of our insured population retains continuous membership for multiple years, some patients may disenroll before a diagnosis or laboratory result is obtained which could result in a potentially reduced incidence estimate. Finally, given that our study was conducted within an integrated healthcare delivery system in Northern California, our findings may not be fully generalizable to other geographic areas, health care systems, or uninsured patients.

In conclusion, we developed a large-scale approach based on EHR data combined with targeted physician adjudication to systematically identify and delineate the incidence and

characteristics of children with primary NS treated in an integrated healthcare delivery system. Further efforts to refine our methods to incorporate more comprehensive EHR data and advanced machine learning methods are needed to promote more efficient population-based identification and characterization of patients with primary NS who may benefit from structured management and follow-up.

## Supporting information

**S1 File.**
(DOCX)

**S1 Data.**
(XLSX)

## Author Contributions

**Conceptualization:** Thida C. Tan, Alan S. Go.

**Data curation:** Rishi V. Parikh, Dongjie Fan, Jingrong Yang.

**Formal analysis:** Dongjie Fan, Jingrong Yang.

**Funding acquisition:** Alan S. Go.

**Investigation:** Alan S. Go.

**Methodology:** David Law, Anne S. Salyer, Leonid Yankulin, Sijie Zheng, Juan D. Ordonez, Glenn M. Chertow, Alan S. Go.

**Project administration:** Thida C. Tan.

**Supervision:** Thida C. Tan.

**Validation:** David Law, Anne S. Salyer, Leonid Yankulin, Sijie Zheng, Juan D. Ordonez, Glenn M. Chertow.

**Visualization:** Rishi V. Parikh.

**Writing – original draft:** Rishi V. Parikh, Farzien Khoshniat-Rad, Alan S. Go.

**Writing – review & editing:** Rishi V. Parikh, Thida C. Tan, David Law, Anne S. Salyer, Leonid Yankulin, Janet M. Wojcicki, Sijie Zheng, Juan D. Ordonez, Glenn M. Chertow, Farzien Khoshniat-Rad, Jingrong Yang, Alan S. Go.

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
