## [Decision Letter · Decision Letter 0]

2 Mar 2021

PONE-D-21-04747

Population-based identification and temporal trend of children with primary nephrotic syndrome: The Kaiser Permanente Nephrotic Syndrome Study

PLOS ONE

Dear Dr. Alan S Go

Thank you for submitting your manuscript to PLOS ONE. After careful consideration, we feel that it has merit but does not fully meet PLOS ONE’s publication criteria as it currently stands. Therefore, we invite you to submit a revised version of the manuscript that addresses the points raised during the review process.

We look forward to receiving your revised manuscript.

Kind regards,

Rajendra Bhimma, PhD

Academic Editor

PLOS ONE

Journal Requirements:

2. In the ethics statement in the manuscript and in the online submission form, please provide additional information about the patient records/samples used in your retrospective study, including: a) whether all data were fully anonymized before you accessed them; b) the date range (month and year) during which patients' medical records/samples were accessed.

"I have read the journal's policy and the authors of this manuscript have the following disclosures: ASG declares funding from the Brin Wojcicki Foundation. GMC declares grants from NIDDK, and Amgen, and fees from Ardelyx, AstraZeneca, Baxter, Cricket, DiaMedica, Gilead, Reata, Sanifit, Vertex, Satellite Healthcare, Angion, Bayer, ReCor and other disclosures from CloudCath, Durect, and Outset. All other authors have nothing to disclose."

Additional Editor Comments:

See comments by reviewers

Reviewers' comments:

Reviewer's Responses to Questions

**Comments to the Author**

1. Is the manuscript technically sound, and do the data support the conclusions?

Reviewer #1: Yes

Reviewer #2: Yes

2. Has the statistical analysis been performed appropriately and rigorously? 

Reviewer #1: Yes

Reviewer #2: I Don't Know

3. Have the authors made all data underlying the findings in their manuscript fully available?

Reviewer #1: Yes

Reviewer #2: Yes

4. Is the manuscript presented in an intelligible fashion and written in standard English?

Reviewer #1: Yes

Reviewer #2: Yes

5. Review Comments to the Author

Reviewer #1: The authors estimate and characterize the population incidence of pediatric primary NS in a large, integrated healthcare delivery system through structured review of available data within electronic health records (EHR) and associated health system data sources. They conclude that this population-based identification of children with primary NS leveraging electronic health records can provide a unique approach and platform for describing the natural history of NS and identifying determinants of outcomes in children with primary NS.

The study is interesting, however, there are some concerns.

Concerns:

1) The supporting literature for the definition of NS in this study should be clearly stated in the text.

2) The indications for renal biopsy in NS in the population in this study should be presented. The indications for renal biopsy may have a significant impact on the data presented in this study.

3) Many pediatric NS patients have MCD, and a significant proportion of these patients do not have renal biopsy. According to the method used by the authors in this study, it should be possible to identify patients with NS who have not undergone renal biopsy. In pediatric patients, the population without renal biopsy is also important and should be included in the analysis.

4) In pediatric NS, the term "idiopathic" is often used. The authors use the term "primary" instead of "idiopathic". This is correct if the disease is confirmed as "primary" by renal biopsy.

On the other hand, the rate of renal biopsy among 179 primary NS is shown to be 40.2%. This statement is strange. How can they distinguish between MCD, FSGS, and MN when no renal biopsy was performed? Does this mean that re-biopsy data was collected in some cases? The authors should clearly explain this point and improve the manuscript to avoid misleading the readers. There is a significant problem with this database if they are diagnosing FSGS and MN without performing renal biopsy.

5) There were 67 cases (37.4%) with missing data for renal function. A total of 67 cases (37.4%) with missing data for renal function is a serious limitation of this study. Why is this the case? Do many attending physicians not measure renal function in NS patients? The same is fact for serum albumin. The authors mention that we additionally performed targeted manual chart review by board-certified nephrologists for evidence of a clinical presentation consistent with NS which enhanced our specificity and reduced misclassification compared with reliance only on administrative diagnosis codes. Also, the authors described that all primary NS cases were manually confirmed through targeted manual medical records review by nephrologists, which is considered the gold standard for retrospective case identification and also demonstrated the suboptimal performance of diagnosis codes alone to identify children with NS. If so, I would guess that kidney function and serum albumin could be easily checked. The high incidence of missing data for renal function and serum albumin in NS studies is fatal. Appropriate improvements in the manuscript are desirable. If this is not possible, then the data extraction itself may be inappropriate, and the credibility of the data presented may be affected.

6) The numbering of the figures is wrong. They do not match the text.

7) In Table 2, the number of kidney biopsies performed is not correct. MCD (30), FSGS (36), and MN (6) add up to only 72, while the total is 74.

Reviewer #2: The authors describe Population-based identification and temporal trend of children with primary nephrotic

syndrome:

1. The study was funded by Brin Wojcicki, so why say ‘Grant Not Applicable’. Please give url of Funder

2. Line 308: What is the percentage of insured versus uninsured population?

6. PLOS authors have the option to publish the peer review history of their article (what does this mean?). If published, this will include your full peer review and any attached files.

Reviewer #1: No

Reviewer #2: No

---

## [Author Response · Author response to Decision Letter 0]

16 Apr 2021

Please see uploaded Response to Reviewers document

---

## [Decision Letter · Decision Letter 1]

14 Jun 2021

PONE-D-21-04747R1

Population-based identification and temporal trend of children with primary nephrotic syndrome: The Kaiser Permanente Nephrotic Syndrome Study

PLOS ONE

Dear Dr. Alan S. Go

Thank you for submitting your manuscript to PLOS ONE. After careful consideration, we feel that it has merit but does not fully meet PLOS ONE’s publication criteria as it currently stands. Therefore, we invite you to submit a revised version of the manuscript that addresses the points raised during the review process.

Please see comments by the reviewers that need to be addressed. 

We look forward to receiving your revised manuscript.

Kind regards,

Rajendra Bhimma, PhD

Academic Editor

PLOS ONE

Additional Editor Comments (if provided):

Please see comments from reviewers

Reviewers' comments:

Reviewer's Responses to Questions

**Comments to the Author**

1. If the authors have adequately addressed your comments raised in a previous round of review and you feel that this manuscript is now acceptable for publication, you may indicate that here to bypass the “Comments to the Author” section, enter your conflict of interest statement in the “Confidential to Editor” section, and submit your "Accept" recommendation.

Reviewer #1: All comments have been addressed

Reviewer #3: All comments have been addressed

Reviewer #4: All comments have been addressed

2. Is the manuscript technically sound, and do the data support the conclusions?

Reviewer #1: Yes

Reviewer #3: Yes

Reviewer #4: Yes

3. Has the statistical analysis been performed appropriately and rigorously? 

Reviewer #1: Yes

Reviewer #3: Yes

Reviewer #4: Yes

4. Have the authors made all data underlying the findings in their manuscript fully available?

Reviewer #1: Yes

Reviewer #3: Yes

Reviewer #4: Yes

5. Is the manuscript presented in an intelligible fashion and written in standard English?

Reviewer #1: Yes

Reviewer #3: Yes

Reviewer #4: Yes

6. Review Comments to the Author

Reviewer #1: (No Response)

Reviewer #3: This is an interesting and well thought out manuscript that uses the electronic health record to assess the incidence of nephrotic syndrome in pediatric patients in a well characterized and excellent health care system.

The following are some suggestions that might improve the manuscript

1. The number of patients is unclear. Table 1 is a bit confusing and it is unclear whether the number of patients is 365 or 179. I think a CONSORT diagram documenting the steps to reach the analytic sample would be helpful

2. The definition of nephrotic syndrome is more appropriate for adults and not pediatric patients where the proteinuria threshold is >2 mg/mg creatinine

3. The impact of race is not clearly presented.

4. The authors might comment on the minor change in the incidence of nephrotic syndrome over time

5. The authors might want to compare their findings to those in adults published in AJKD that was based on the same health care system.

Reviewer #4: To be accepted for publication in PLOS ONE, research articles must satisfy the following criteria:

1. The study presents the results of original research.

Nephrotic syndrome (NS) is a clinical diagnosis with proteinuria, hypoalbuminemia and odema. Although primary NS is relatively uncommon, it accounts for mortality and considerable morbidity in young children. The prognosis of NS in children correlates with the spectrum of responsiveness to steroid therapy, from steroid-sensitive NS (SSNS) to steroid-resistant NS (SRNS). SRNS is the most common acquired cause of end-stage renal disease (ESRD) in children.

Kaiser Permanente (KP) in the Northern California region, provides an integrated healthcare delivery system caring for >750,000 children. Primary NS in the region is relatively uncommon, and no prior population-based research in the region has been carried out to study the incidence of primary NS in children. The incidence being unknown, both based on the clinical patterns and ethnic distribution despite the data being available in the EMRs and laboratory databases of the KP systems.

The aim of this study was to estimate the incidence of primary NS in the region based on a retrospective laboratory and EMR data with a multicenter collaboration within the KP systems.

Such data would help in improving population-level quality of care and thereby clinical outcomes. Additionally, would contribute to the development and implementation of system-wide efforts to improve complex delivery care systems for the illness. This research data despite its limitations of being retrospective and only 40% of the children having a renal biopsy, will facilitate infrastructure and processes that support the ability to accelerate large-scale change internally, as well as externally through successful multi-institutional collaborations within the Kaiser Permanente as seen in the paper. In conclusion the study does present results of original research work done in the region of northern California cared for by the Kaiser Permanente integrated healthcare delivery system and agree with the authors that the study can provide a unique approach and platform for describing the natural history of NS and identifying determinants of outcomes in children with primary NS.

2. Results reported have not been published elsewhere.

The results reported have not been published elsewhere as far as my search for the same has revealed. This data although retrospective does provide a comphrensive input into the incidence of pediatric nephrotic syndrome in the region split by clinical presentation and ethnic breakup.

3. Experiments, statistics, and other analyses are performed to a high technical standard and are described in sufficient detail.

The analysis is preformed to the needed technical standard and done in sufficent detail for the data that is available. All data are made fully available without restriction and the authors have made available a file of the minimal de-identified data set necessary to replicate the analyses in this manuscript.

4. Conclusions are presented in an appropriate fashion and are supported by the data.

The data is retrospective and collected from the EMRs as well as laboratory databases, from across the KP organization. There could be changes in the etiology and the racial presentation presented, since only 40% have biopsy data and there is no long-term follow-up reported after the initial diagnosis. Especially, that in 60% of the patients the diagnosis was entirely clinical and not supported with renal histology. Although ethnic differences in clinical presentations are shown, it would have been useful to have follow-up data on outcomes among these children with nephrotic syndrome which are unknown. Despite the higher incidence of NS among ethnics, outcomes may be significantly less complicated when compared between various clinical presentations. Within the above limitations the presentation is appropriately supported.

5. The article is presented in an intelligible fashion and is written in standard English.

Yes, the article is presented in an intelligible fashion and written in standard English with no flaws observed within the article.

6. The research meets all applicable standards for the ethics of experimentation and research integrity.

The research does meet with all the applicable standards for the ethics of experimentation and research integrity. The study was approved by the KPNC institutional review board, and waiver of informed consent was obtained due to the nature of this retrospective data-only study, with the use of de-identified data sets as reported.

The authors have also made the relevant disclosures which do not conflict with the results of the study: ASG declares funding from the Brin Wojcicki Foundation. GMC declares grants from NIDDK, and Amgen, and fees from Ardelyx, AstraZeneca, Baxter, Cricket, DiaMedica, Gilead, Reata, Sanifit, Vertex, Satellite Healthcare, Angion, Bayer, ReCor and other disclosures from CloudCath, Durect, and Outset. All other authors have nothing to disclose.

7. The article adheres to appropriate reporting guidelines and community standards for data availability.

The manuscript does provide a clear and complete account of the research done. The reporting does clearly allow editors, peer reviewers and readers to understand what was done and how. The data can be clearly understood by a reader, replicated by a researcher, and used to make a clinical decision and included in a systematic reveiw with its limitations.

7. PLOS authors have the option to publish the peer review history of their article (what does this mean?). If published, this will include your full peer review and any attached files.

Reviewer #1: No

Reviewer #3: **Yes: **Howard Trachtman

Reviewer #4: No

---

## [Author Response · Author response to Decision Letter 1]

24 Jun 2021

We thank the Editors and Reviewers for providing comments on the manuscript, and have provided a point-by-point response to comments in the attached file.

---

## [Editor Report · Decision Letter 2]

8 Sep 2021

Population-based identification and temporal trend of children with primary nephrotic syndrome: The Kaiser Permanente Nephrotic Syndrome Study

PONE-D-21-04747R2

Dear Dr. Alan S. Go

We’re pleased to inform you that your manuscript has been judged scientifically suitable for publication and will be formally accepted for publication once it meets all outstanding technical requirements.

Kind regards,

Rajendra Bhimma, PhD

Academic Editor

PLOS ONE

Additional Editor Comments (optional):

Dear Dr Go

Thank you for your submission. I apologies for the delay but now that i have received the reviewers comments, I am happy to forward this to to the Editor in Chief for a final decision on the manuscript.
---

## [Editor Report · Acceptance letter]

6 Oct 2021

PONE-D-21-04747R2 

Population-based identification and temporal trend of children with primary nephrotic syndrome: The Kaiser Permanente Nephrotic Syndrome Study 

Dear Dr. Go:

I'm pleased to inform you that your manuscript has been deemed suitable for publication in PLOS ONE. Congratulations! Your manuscript is now with our production department. 

Kind regards, 

on behalf of

Professor Rajendra Bhimma 

Academic Editor

PLOS ONE